# Oxidation Resistance of a Si–TiSi_2_–MoSi_2_–TiB_2_–CaSi_2_ Coating on a C_f_/C–SiC Substrate in High-Speed High-Enthalpy Air Plasma Flows

**DOI:** 10.3390/nano11102637

**Published:** 2021-10-07

**Authors:** Alexey Astapov, Lev Rabinskiy, Olga Tushavina

**Affiliations:** 1General Engineering Training Institute, Moscow Aviation Institute, National Research University, 125993 Moscow, Russia; rabinskiy@mail.ru; 2Aerospace Institute, Moscow Aviation Institute, National Research University, 125993 Moscow, Russia; solgtu@gmail.com

**Keywords:** heat-resistant coating, C_f_/C–SiC composite, oxidation, heterogeneous film, kinetics, air plasma, fire tests, catalyticity

## Abstract

The results of a study on the development and testing of a heat-resistant coating in a Si–TiSi_2_–MoSi_2_–TiB_2_–CaSi_2_ system to protect C_f_/C–SiC composites from oxidation and erosional entrainment in high-speed flows are presented here. The coating was formed using firing fusion technology on the powder composition. Oxidation resistance tests were carried out under static conditions in air at 1650 °C and under conditions of interaction with high-speed air plasma flows, with Mach numbers M = 5.5–6.0 and enthalpy 40–50 MJ/kg. The effectiveness of the protective action of the coating was confirmed at surface temperatures of *T_w_* = 1810–1820 °C for at least 920–930 s, at *T_w_* = 1850–1860 °C for not less than 510–520 s, at *T_w_* = 1900–1920 °C for not less than 280–290 s, and at *T_w_* = 1940–1960 °C for not less than 100–110 s. The values of the rate of loss of the coating mass and the rate constant of heterogeneous recombination of atoms and ions of air plasma on its surface were estimated. The performance of the coating was ensured by the structural-phase state of its main layer, and the formation and evolution on its surface during operation of a passivating heterogeneous oxide film. This film, in turn, is composed of borosilicate glass with titanium and calcium liquation inhomogeneities, reinforcing TiO_2_ microneedles and in situ Si_2_ON_2_ fibers. It was shown that at *T_w_* ≥ 1850–1860 °C, the generation of volatile silicon monoxide was observed at the “oxide layer–coating” interface, followed by the effects of boiling and breakdown degradation of the oxide film, which significantly reduced the lifespan of the protective action of the coating.

## 1. Introduction

Currently, carbon–carbon and carbon–ceramic composites are the most promising materials for use in thermal protection systems for airframes and flow paths of propulsion systems of atmospheric high-speed aircraft and reusable aerospace vehicles [1,2,3,4,5,6]. Their main advantages include low density, low coefficients of thermal expansion, and high specific mechanical characteristics up to 2500 °C, including fracture toughness and impact toughness. However, in oxygen-containing media, their use is limited by the tendency of carbon to oxidize, starting from temperatures of 400–450 °C, and insufficient heat-resistance of ceramic matrices, which leads to their destruction. In high-speed high-enthalpy flows, the degradation of the structure of composites is significantly aggravated as a result of the simultaneous occurrence of thermochemical processes (oxidation, heterogeneous recombination), erosion, and ablation. The most effective way to increase the working temperatures of carbon-based materials is to apply heat-resistant anti-ablative coatings on surfaces in contact with oxidizing media [3,6,7,8,9,10]. Increasing requirements for promising heat-shielding systems requires the creation of new, more effective coatings in comparison with existing technical solutions.

When designing coatings for extreme operating conditions, the concept of self-healing has become widespread [9,11,12,13], based on the use in the structure of coatings of a glass phase or components capable of glass formation during operation. The glass phase encapsulates the underlying layers and heals microdefects, ensuring the operability of the composition until the nominal supply of glass-forming components is exhausted. The introduction of a relatively low-melting non-oxide phase into the composition of the coatings, which is in a liquid or viscous-flowing state at operating temperatures, increases their ability to heal defects, accelerates the formation of a protective oxide film, and facilitates the removal of gaseous products inevitably formed during the oxidation process. For example, heat-resistant coatings based on the systems SiC–MoSi_2_–CrSi_2_ [14,15], MoSi_2_–CrSi_2_––Si/SiC [16,17,18], MoSi_2_–CrSi_2_–Si/B-modified SiC [19], SiC–CrSi_2_–Cr_3_C_2_–MoSi_2_-Mo_2_C [20] and CrSi_2_–ZrSi_2_–Y_2_O_3_/SiC [21] contain the CrSi_2_ phase (*T_melt_* = 1490 °C), which heals microcracks and reduces the porosity of the coatings. In addition, the oxide Cr_2_O_3_ (*T_melt_* = 2435 °C) formed during oxidation acts as a “fixing” phase, preventing the formation and propagation of microcracks in the oxide film (pinning effect). The performance of the coatings has been confirmed under static oxidation conditions in air at 1500 °C [14,20,21] and 1600 °C [15,16,17,19], as well as in environments with an increased oxidation potential (50% H_2_O–50%O_2_, 50% O_2_–50% Ar) at 1500 °C [18].

The role of the healing phase in the inner layers of coatings of the MoSi_2_–Mo_5_Si_3_–HfSi_2_/TaSi_2_/TaC and WSi_2_–W_5_Si_3_–HfSi_2_/TaSi_2_/TaC systems [22], Si–TiSi_2_–MoSi_2_–TiB_2_ [11,23], Si–TiSi_2_–MoSi_2_–TiB_2_–SiC_w_ [24], ZrSi_2_/Si–ZrSi_2_/SiC [25] and ZrSi_2_–MoSi_2_–ZrB_2_ [13,26] perform eutectics between silicides [22] or between silicon and silicides [11,13,23,24,25,26]. The presence of eutectics between components in the composition of the coatings improves the adhesion between the layers and to the substrate, accelerates the regeneration of the oxide film, and also to some extent compensates for the differences in the coefficients of thermal expansion of coatings and protected materials. The effectiveness of the protective action of coatings was confirmed under static oxidation conditions in air at 1650 °C [13,25], in an oxidizing gas flow at 1900 °C [22], in high-speed high-enthalpy air plasma flows at 1850 °C [11,24], 1950 °C [23], 2000 °C [26], and 2200 °C [13], as well as under the conditions of an ablation fire experiment at 2400 °C [25].

Special attention should be paid to heat-resistant self-healing coatings based on the Si–TiSi_2_–MoSi_2_–TiB_2_ system, including [11,23,27], wt%: titanium 15–40, molybdenum 5–30, boron 0.5–2.5 and silicon—the remainder. In the process of obtaining coatings, a dendritic-cellular structure is formed from disilicide phases (Ti_x_Mo_1−x_Si_2_ (0.1 < x < 0.87), TiSi_2_, MoSi_2_). The space inside is filled with low-melting (*T_melt_* ~ 1320 °C) eutectic (Si + Ti_x_Mo_1−x_Si_2_ + TiSi_2_)_e_ and refractory particles TiB_2_ (*T_melt_* = 2970 °C). Oxidation transforms the coating into a multilayer structure with the formation of a heterogeneous oxide film. The self-healing effect consists of the rapid filling of random defects with a eutectic and accelerated, in comparison with known coatings, formation of a protective oxide film through the liquid phase (eutectic). Resistance to erosion entrainment is ensured by the presence of a branched silicide framework (skeleton), additionally reinforced with refractory TiB_2_ particles.

Coatings provide effective protection of C_f_/C and C_f_/SiC composites from oxidation and erosion up to temperatures on the surface *T_w_* ~ 1800–1820 °C under conditions of air–gas dynamic flow and nonequilibrium heating by air plasma flows with Mach numbers M = 5–6 and enthalpy 35–45 MJ/kg. The coatings have a low catalytic activity (the rate constant of heterogeneous recombination of atoms and ions at a flux *K_w_* = 3–5 m/s) and satisfactory characteristics of emissivity (the degree of blackness of the total normal radiation ε ≈ 0.7). Degradation of the structure of coatings, and, consequently, the loss of their operational properties, occurs mainly for two reasons [23]: Firstly, as the result of increases in erosional entrainment with increases in temperature; this is due to decreases in the number of skeleton-forming silicide phases due to their partial dissolution in the liquid phase upon reaching the solidus temperatures. Secondly, as a result of discontinuity of the oxide film due to formation at the “oxide film-coating” interface and the outflow of gaseous oxidation products (mainly SiO) under conditions of significant external rarefaction (*P_w_* ≤ 0.1 atm.). The rates of erosional entrainment and sublimation increase with increasing temperatures and with decreasing ambient pressure.

The need to increase the operating temperatures of the class of coatings under consideration has led to research aimed at increasing the refractoriness of their main (unoxidized) layer and the viscosity of oxide films formed during the oxidation process. In our opinion, promising directions for improving the chemical and phase compositions of coatings are increases in the content of the refractory TiB_2_ phase in their structure and the introduction of calcium into the composition of the coatings, which is a surfactant element that is supposed to be combined with other components; this will increase the degree of heterogeneity of the formed oxide film and improve its functional properties.

The aim of this study was to obtain a heat-resistant coating of an experimental composition in the Si–TiSi_2_–MoSi_2_–TiB_2_–CaSi_2_ system on a C_f_/C–SiC composite and to study its oxidation resistance under conditions of interaction with high-speed high-enthalpy air plasma flows at operating temperatures on the surface of up to *T_w_* = 1900–1950 °C.

## 2. Materials and Methods

Powders of KR00 technical silicon with a purity of 99.0%, PTOM-1 titanium with a purity of 99.9%, PM-99.95 molybdenum with a purity of 99.95%, B-99A amorphous boron with a purity of 99.0%, and TU 95.768-80 metal calcium with a purity of 98.5% were used as the initial alloy components for the coating. The powders were mixed in a rotary ball mill in steel drums using carbide grinding bodies. To prevent self-ignition of titanium powder, mixing was carried out in isopropyl alcohol. The finished mixture was dried in a vacuum drying cabinet at a temperature of 80 °C and then pressed into cylindrical specimens 18 mm in diameter and 15 mm in height on a ZDM 50E hydraulic press (VEB WPM Leipzig, Markkleeberg, Germany) with a force of 25 t. Melting was carried out in suspension in an inert atmosphere of a crucible-free induction furnace equipped with a high-frequency electromagnetic inductor. High-purity helium (grade 7.0) was used as an inert gas. The alloy was cast into copper molds with a diameter of 10 mm. To eliminate segregation, the ingots were annealed in a shaft-type vacuum furnace SSHVE-1.2.5/25 I2 (LLC “OZ VNIIETO”, Moscow, Russia) at a temperature of 1100 °C for 5 h at a residual gas pressure in the chamber of 5–6 MPa.

The powder was prepared from the melted ingots by crushing them on the same press with an effort of 14–15 tons, followed by grinding in a rotating ball mill with steel balls with a hardness of HRC = 58–62 to a fineness of ~ 6–8 microns. To improve the flowability of the powder, and reduce the amount of bound and adsorbed moisture as well as organic contaminants, the powder material was calcined in air during 5 h at 200 °C.

The coating was formed by the method of firing layers from the obtained powder, supplemented with KR00 silicon in a mass ratio of 4:1. Extra-blending with silicon was carried out in order to reduce the firing temperature and increase the continuity of the coating structure. As a substrate, we used samples in the form of disks with a size of Ø 30 mm × 8.5 mm, made of a carbon–ceramic composite material of the C_f_/C–SiC class based on TKK-2 carbon fabric and a combined matrix of phenolic resin coke, pyrolytic carbon, and silicon carbide. The composite was characterized by a volumetric ratio of C:SiC phases equal to 4:1, a density of 1.85 g/cm^3^, a flexural strength of 100 MPa, and a thermal conductivity of 35 W/(m·K). Heat treatment was carried out to a temperature of 1480 ± 5 °C in the same vacuum furnace at a residual pressure of 8–9 MPa. The temperature was measured with a W–Re type A1 thermocouple. Before testing for oxidation resistance, the samples were subjected to ultrasonic cleaning in ethanol on an UZV-5.7 unit (LLC “Sapfir”, Moscow, Russia).

Compact samples for studying oxidation resistance under static conditions were obtained by hot pressing the synthesized powder. The process was carried out on a DSP-515 SA unit (Dr. Fritsch, Fellbach, Germany) in vacuum at a temperature of 1250 ± 5 °C, a heating rate of 50 °C/min, a pressure of 30 MPa, and isobaric holding for 10 min. The resulting compacts in the form of disks 50 mm in diameter and 5 mm thick were cut into samples for experimental studies in the form of square plates 10 mm × 10 mm × 5 mm in size using an ARTA 200-2 two-coordinate EDM machine (NPK “Delta-Test”, Fryazino, Russia). Before investigations, the edges of the samples were polished on a TegraPol-11 installation (Struers, Ballerup, Denmark), followed by ultrasonic cleaning in ethanol in the same installation as for cleaning the coatings.

The compacts were oxidized in an LHT 04/17 SW chamber electric furnace (Nabertherm, Lilienthal, Germany) with a working space of 4 L in air. A type B Pt–Rh thermocouple was used as a temperature sensor. Samples in corundum crucibles, preliminarily annealed to a constant weight, were loaded into a furnace at a chamber temperature of 20 °C, then heated to 1650 °C at a heating rate of ~ 40 °C/min, then kept isothermal for 60 min, cooled together with the furnace to 20 °C (the time of temperature decrease in the intervals 1650–1250 °C, 1250–750 °C, 750–250 °C and 250–20 °C was 8, 40, 120 and 160 min, respectively), removed from the oven and weighed. The total number of thermal cycles was 5, i.e., the total time of isothermal holding at 1650 °C was 5 h. For each composition, 4 samples were tested. The samples were weighed on a GR-202 analytical balance (AND, Tokyo, Japan) with an accuracy of 10^−4^ g. The experimental data on the oxidation kinetics were approximated by the least squares method using the Levenberg–Marquardt algorithm, using a mathematical package included in the MathCAD software. The accuracy of the approximation was assessed through the Pearson correlation coefficient *r*.

Gas-dynamic bench tests were carried out on an NIO-8 FSUE TsAGI (Zhukovsky, Russia) on a VAT-104 aerodynamic stand equipped with an induction plasmatron for gas heating. The equipment and methods for conducting fire experiments are described in detail in [24,28]. The processes of thermochemical interaction of samples with high-speed flows of air plasma were simulated (for the conditions of flight of promising reentry vehicles in the Earth’s atmosphere at an altitude of 80–100 km). The air plasma parameters were within the following limits: flow velocity 4.3–4.8 km/s (Mach number M = 5.5–6.0); enthalpy of flow 40–50 MJ/kg; flow stagnation temperature ~ 9000–10,000 K; gas pressure in front of the samples 1.0–3.8 kPa; the degree of air dissociation in the flow was 80–90%; the degree of ionization was about 1%. The outlet section of the nozzle had a diameter of 53.7 mm, the diameter of the underexpanded plasma jet was ~ 100 mm, and the distance from the nozzle exit to the Mach disk was 250 mm. An increase in the average surface temperature of the samples during the firing experiments was carried out by a stepwise increase in pressure in the preheater prechamber *P*_0_. The temperatures at the front and back surfaces of the samples, *T_w_*, reached during the tests were measured with a VS-CTT-285/E/P-2001 pyrometer at a wavelength of 890 nm, taking into account correction for the spectral emissivity of the coating, which was taken equal to ε = 0.7. The samples were weighed on the same balance as in static oxidation.

The rate constant of heterogeneous recombination of atoms *K_w_* at the active centers of the coating surface was determined from the difference in the heat flux density to the reference and investigated coatings. Using parametric numerical simulation of the flow and heat transfer of the sample, the derivative *dK_w_/dT_w_* was calculated [29]. The *K_w_* value for the samples under study was found from the known *K_ws_* value for the reference sample, value *dK_w_/dT_w_* and temperature differences ∆*T_w_* heat-insulated test and reference samples [29]:*K_w_* = *K_ws_* + (*dK_w_*/*dT_w_*)·∆*T_w_*.(1)

As a reference, we used samples of a C_f_/C–SiC composite with a base coating in the Si–TiSi_2_–MoSi_2_–TiB_2_ system [11,27], for which the rate constant of heterogeneous recombination was determined earlier [30].

The study of the chemical composition of the obtained powder was carried out by X-ray fluorescence analysis on an ARL OPTIM’X wave spectrometer (Thermo Fisher Scientific, Waltham, MA, USA) with the possibility of determining elements in the range from fluorine to uranium and the absence of the possibility of analyzing light elements (boron, carbon, oxygen).

X-ray diffraction patterns were taken on an ARL X’tra diffractometer (Thermo Fisher Scientific, Switzerland) with a Cu *K*α copper anode. Horizontal collimation slits limiting the vertical divergence of the X-ray beam were selected depending on the size of the areas under study, varying them in the range from 0.5 to 1.5 mm. To register the radiation, a Si (Li) semiconductor detector tuned to the Cu *K*α 1,2 doublet was used. X-ray diffraction patterns were taken with a step of 0.02° at a goniometer radius of 520 mm at a rate of 0.5°/min. X-ray phase analysis (XRD) was performed using the Crystallographica Search-Match V.3,1,0,0 program (Oxford Cryosystems, Oxford, UK) and the ICDD PDF-2 database of reference radiographs (2010). To assess the quantitative content of the phases, we used the corundum number method and/or the Rietveld method using SIROQUANT V3 software (Sietronics Pty Ltd, Canberra., Australia).

Microstructural studies were performed using scanning electron microscopes (SEM) EVO-40 (Carl Zeiss, Oberkochen, Germany) and Phenom XL (Phenom-World BV, Eindhoven, The Netherlands) equipped with X-ray energy dispersive spectrometers (EDS). The elemental composition was determined at an accelerating voltage of 10–15 kV and a probe current of 1 nA. Precision equipment from Struers (Ballerup, Denmark) was used to prepare metallographic sections.

The adhesive strength of the coating was determined in accordance with GOST 32299-2013 on a universal testing machine LFM-300 (Walter + Bai AG, Löhningen, Switzerland) according to the method of destruction of glued samples at normal separation. Bonding was carried out according to OST 92-0949-74 using cold-cured epoxy-polyamide adhesive VK-9.

The isobaric–isothermal potential ΔG (Gibbs free energy) of possible chemical reactions during oxidation was calculated using the online calculator FACT developed at Ecole Polytechnique and McGill University (Montreal, QC, Canada) [31].

## 3. Results and Discussion

### 3.1. Selection of the Experimental Coating Composition

The choice of coating composition for experimental studies was due to the following considerations: With an increase in boron content, the number of dispersed particles of titanium diboride TiB_2_ in the coating volume will increase. Their refractoriness and high thermodynamic stability (in comparison with frame-forming disilicides) should contribute to an increase in the temperature resistance of the main coating layer, which will have a positive effect on its resistance to erosion entrainment. From this position, it is advisable to introduce the maximum possible amount of boron. However, based on the results of studies on additional modification of coatings of the Si–TiSi_2_–MoSi_2_–TiB_2_ system with SiC_w_ whiskers [24], with a significant content of TiB_2_ particles, an increase in the rate of oxidation of coatings should be expected due to a decrease in the amount of eutectic in their structure. For coatings of the Si–TiSi_2_–MoSi_2_–TiB_2_–SiC_w_ system, this problem occurred when SiC_w_ was added in excess of 15 wt%.

Along with this, an increase in the boron content should have a positive effect on the functional properties of the formed surface oxide film. Boron oxide B_2_O_3_ reduces the crystallization ability of silicon dioxide SiO_2_ as a result of decreases in the structuring of borosilicate glass SiO_2_·B_2_O_3_ formed during fusion. The amorphization of the oxide film, in turn, decreases its gas permeability and reduces the probability of recombination of surface flow atoms and ions, i.e., increases anti-catalytic properties. A decrease in the solidus temperature and viscosity of borosilicate glass (in comparison with silica) contributes to an increase in its wetting properties, which leads to an improvement in its ability to self-heal defects. However, at the same time, the resistance of the oxide film to mechanical entrainment or runoff in high-speed gas flows will decrease.

On the other hand, it is known that borosilicate glasses containing oxides of transition metals of groups IV–VI of D.I. Mendeleev’s periodic system of chemical elements, including TiO_2_ and MoO_3_, have a strong tendency to phase separation (immiscibility) [32,33,34]. Heterogeneous oxide systems are characterized by an increase in liquidus temperatures and increased viscosity. In turn, an increase in viscosity decreases the rate of oxygen diffusion through the oxide film according to the Stokes–Einstein relation. Another potential advantage of increased viscosity along with an increased liquidus temperature is a slight decrease in the vapor pressure of heterogeneous borosilicate films as compared to homogeneous borosilicate glass [35,36]. However, it should be taken into account that the catalytic activity of heterogeneous silicate systems will be higher than that of similar homogeneous ones [13,37].

During oxidation of coatings in the Si–TiSi_2_–MoSi_2_–TiB_2_ system, a heterogeneous oxide film is formed, consisting of borosilicate glass SiO_2_·B_2_O_3_ and titanium oxide TiO_2_ immiscible with it in the form of rutile [11,23]. The latter is a vast network of microscopic needles, located absolutely randomly, which leads to the appearance of the effect of “reinforcement” of the oxide film, and, consequently, to an additional increase in the resistance to erosion entrainment in flows.

In order to increase the degree of heterogeneity of the oxide film, this work proposes an additional introduction of calcium into the coating composition. The decisive factor in the choice of calcium was, on the one hand, its surface activity with respect to silicon and reactivity when interacting with oxygen, and, on the other hand, the ability of its cations (Ca^2+^) to act as a modifier and lead to the formation of microheterogeneous regions in silicate glasses [33,34,38,39]. An increase in the amount of these cations should contribute to a decrease in the degree of polymerization of the structural framework of modified glasses, which will be expressed in their greater ability to segregate (liquid immiscibility) and, ultimately, lead to delamination.

Taking into account the above reasoning and performed preliminary experimental studies [40], the object of study in this work is a coating in the Si–TiSi_2_–MoSi_2_–TiB_2_–CaSi_2_ system, containing boron and calcium within 10 and 5 wt%, respectively.

### 3.2. Chemical and Phase Composition of the Obtained Powder and Coating

The chemical composition of the obtained heterophase powder (without taking boron into account), was wt%: Si = 44.2, Ti = 28.4, Mo = 17.9, Ca = 6.6, Fe = 0.5, and the rest consisted of impurities It should be noted that since boron is included in the form of titanium diboride TiB_2_, it was impossible to determine the exact content of titanium in a sample by this method—the data obtained can only be regarded as indicative. The iron found in the powder should be interpreted as a result of grinding when dispersing with steel balls.

Table 1 shows the phase composition of the obtained powder and the coating formed from it. The main phases in the powder were titanium diboride TiB_2_ and titanium and molybdenum disilicides Ti_0_._8_Mo_0_._2_Si_2_, TiSi_2_, MoSi_2_, as well as calcium disilicides CaSi_2_ of tetragonal and rhombohedral modifications. In addition to these compounds, the coating contained a significant amount of elemental silicon (the result of additional charging of the powder with silicon during the formation of the coating) and a small number of dispersed particles of silicon carbide SiC located at the boundaries of silicon grains that were hardly distinguishable on microstructures. The latter are formed as a result of the physicochemical interaction of silicon with carbon according to the dissolution–precipitation mechanism [41,42]. Carbon is formed as a result of pyrolysis of the organic binder during heat treatment, and also diffuses from the C_f_/C–SiC substrate.

A typical structure of the cross-section of the coating in the initial state is shown in Figure 1 respectively, in the form of an image of the microstructure in secondary electrons, separate X-ray maps of the distribution of elements and a multilayer combined image created on the basis of combining an electronic photograph and X-ray maps. It can be seen that the disilicide phases were skeletal; the silicon-based eutectic (Si + Ti_x_Mo_1−x_Si_2_ + TiSi_2_)_e_ was inside a kind of dendritic-cellular framework formed by disilicides. TiB_2_ particles were represented by crystals of regular cut, evenly distributed in the coating volume. The average particle size of TiB_2_ did not exceed 3–4 microns. The coating thickness was in the range of 50–60 microns (depending on the relief of the surface of the C_f_/C–SiC composite samples).

The lower limit of the adhesive bond strength of the coating with the substrate was ~20 MPa (separation along the adhesive joint).

### 3.3. Kinetics of Compact Oxidation in Air at 1650 °C

To confirm the correctness of the chosen direction for improving the coatings in the system under consideration, tests for the oxidation resistance of the samples were carried out under static oxidation conditions in air. The samples were compacts obtained by hot pressing of the synthesized powder in the Si–TiSi_2_–MoSi_2_–TiB_2_–CaSi_2_ system. For comparison, we used compacts obtained in a similar manner from a powder in the basic system Si–TiSi_2_–MoSi_2_–TiB_2_ [11,27].

The kinetic curves of compacts oxidation at 1650 °C are shown in Figure 2a in the form of dependences of the weight gain per unit surface area of the samples *q* on test time *t*. Here, markers show the experimental data, and the solid curves show the results of their approximation using a power-law dependence:*q^n^* = *k*·*t*,(2)
where *k* is the oxidation constant; *n* is the exponent. Index *n* reflects the cumulative effect on the oxidation kinetics of such factors as: (i) decreases in the rate of diffusion of reagents through the oxide layer as a result of increases in its thickness (diffusion interpretation of the oxidation mechanism, *n* = 2); (ii) increases in the rate of redox reactions due to partial disruption of the film continuity due to the occurrence of significant stresses on it (diffusion-kinetic mechanism of oxidation, 1 < *n* < 2); (iii) deceleration of the growth of the oxide layer as a result of structural and phase changes in the film, additional inhibition of the diffusion of reagents due to the action of compressive stresses, etc. (oxidation complicated by secondary factors, *n* > 2). The kinetics of the oxidation of films with higher heat resistance is described by the power law with a higher value of *n*. The physical significance of the parameters *k* and *n* is more fully covered in [43].

The values of the parameters *k, n* and the Pearson correlation coefficient *r* obtained by approximating the experimental data for each composition are presented in Table 2. Experimental data on the specific weight gain of samples *q_Σ_* after 5 h of isothermal holding at 1650 °C are also given there. It can be seen that the kinetics of oxidation of compositions *1* and *2* were reliably described (*r* ~ 1) by a power law with an exponent *n* = 2.695 and 1.286, respectively. The value of the exponent *n* > 2 indicated evolutionary changes in the structure of the oxide film formed on compacts of composition *1*, leading to an additional increase in heat resistance. This correlates well with the final specific weight gain *q_Σ_* of composition *1* in comparison with composition *2*.

Figure 2b shows the time variation of the oxidation rates of the samples. It can be seen that at the initial moment of oxidation, the rates were at maximum; however, due to the rapid formation of heat-resistant films, their values rapidly decreased (stage I). Subsequently, the oxidation rates continued to decrease monotonically (stage II) due to the continuous inhibition of the diffusion of reagents through the growing oxide layer, and their values essentially reached a plateau. As a boundary criterion dividing these stages, we, by analogy with the half-life of quantum mechanical systems, chose the time *t*_1/2_, during which the level of the specific weight gain of the samples will be ½ of their total weight gain *q*_Σ_ for the entire oxidation time. The calculated values of the indicated times for each composition are given in Table 2. The same figure shows the calculated values of the average oxidation rates *v_av_* of the samples at each of the stages. It can be seen that at stage I the average oxidation rate *v_av_* of composition *1* was 1.12 times higher than that of composition *2*, which led to an earlier transition of compacts *1* (*t*_1/2_ = 46.3 min versus 123.1 min for compacts *2*) to the stage of delayed oxidation (stage II). It should also be noted that at stage II the *v_av_* index for composition *1* was an order of magnitude lower than at stage I, due to the formation of a heat-resistant oxide layer.

A feature of the performed experiments on oxidation with isothermal holding at a temperature of 1650 °C was the presence of low-temperature stages associated with heating and cooling of compacts together with the furnace (see Section 2). In this case, at relatively low temperatures (~750–1250 °C), an unfavorable oxidation mode is inevitably implemented, where the temperature is not sufficient for the formation of a continuous glass-like layer at the heating stage in the first test cycle or its transition to a viscous-plastic state at subsequent thermal cycling. During thermal cycling in the oxide layer at the cooling stage, as a result of relaxation of tensile stresses arising from the difference in the coefficients of thermal expansion of compacts and formed on the surface of oxide layers, pores and microcracks are formed [43], through which oxygen actively penetrates into the samples. This led to additional deep oxidation of the samples at low-temperature stages of testing, the total duration of which was ~5 h, i.e., as much as the total high-temperature isothermal holding. Therefore, the obtained kinetic regularities integrally reflect the behavior of the samples during thermal cycling in the range of 20–1650 °C, including the taking into account of their behavior under conditions where the protective properties of oxide films cannot be fully realized.

Figure 3 shows the typical results of structural studies (according to SEM and EDX) of an oxide film formed on the surface of compacts in the Si–TiSi_2_–MoSi_2_–TiB_2_–CaSi_2_ system after 5 h oxidation at 1650 °C in air. It can be seen that the oxide film had a complex heterogeneous structure and a significant thickness of ~0.9–1 mm. The film was mainly represented by two immiscible components—a glass phase, which was close in elemental composition to SiO_2_, and solutions in the CaSi_2_O_5_–CaTiSiO_5_ system. The oxide layer contained rounded grains derived from the unoxidized Ti_x_Mo_1−x_Si_2_ phase with sizes of 1 to 80 μm and dispersed TiB_2_ particles with an average size not exceeding 3–4 μm. Increases in the linear dimensions of individual Ti_x_Mo_1−x_Si_2_ grains in comparison with their size in the initial powder and compacts is associated with the recrystallization processes characteristic of liquid-phase sintering. When testing for oxidation resistance, along with the formation of an oxide film, small Ti_x_Mo_1−x_Si_2_ particles dissolve in the melt represented by Si and the eutectic (Si + Ti_x_Mo_1−x_Si_2_ + TiSi_2_)_e_, and their subsequent crystallization on the surface of large particles. As a result, the larger Ti_x_Mo_1−x_Si_2_ grains grow at the expense of the small ones, assuming a rounded shape, which ensures a decrease in their surface energy. The high thickness of the oxide film was associated with a significant fraction of liquid phase in the compact structure at the test temperature. As is known, the liquid phase provides faster mass transfer of reagents, which accelerates the redox processes [13,43,44]. An unoxidized compact was located under the film, represented by the same phases as in the initial state.

During the oxidation of compacts in the basic system Si–TiSi_2_–MoSi_2_–TiB_2_, a heterogeneous oxide film was formed, consisting of a glass phase based on SiO_2_ and titanium oxide TiO_2_ immiscible with it. The oxide layer also contained encapsulated Ti_x_Mo_1−x_Si_2_ and TiB_2_ phases. The layer thickness was 1.5–1.6 times greater than on compacts in the Si–TiSi_2_–MoSi_2_–TiB_2_–CaSi_2_ system, which correlates well with the data on the oxidation kinetics (Figure 2, Table 2). Thus, the results obtained indicate the correctness of the chosen direction for improving the coatings in the system under consideration.

### 3.4. Coating Behavior under Conditions of Heating by a High-Speed Air Plasma Flow

The first series of gas-dynamic tests was carried out with the arrangement of samples from a C_f_/C–SiC composite with the studied coating in the Si–TiSi_2_–MoSi_2_–TiB_2_–CaSi_2_ system at a distance of 70 mm from the outlet section of the plasmatron *W_a_* = 232 ± 4 kW. The air plasma parameters were within the following limits: flow velocity 4.3–4.5 km/s; enthalpy of flow 40–45 MJ/kg; flow stagnation temperature ~ 9000 K; the degree of air dissociation in the flow was 80–85%; the degree of ionization was about 1%.

Three samples were tested under conditions of flow and non-equilibrium heating by an air plasma flow with a fixed temperature level on the surface *T_w_* = 1810–1820 °C for 920–930 s. Typical parameters of the firing experiments mode are shown in Figure 4a. A typical appearance of the front surface of the samples after testing is shown in Figure 4b. The coating on all three samples remained intact. The color of the coating was light gray, in the vicinity of the critical point (the epicenter of the impact of the flow), with a bluish tint and vitreous luster. The average rate of mass loss by the samples for the entire time of the firing experiment in this regime was 7.4 ± 0.8 mg/(cm^2^·h).

The other three samples were tested under similar conditions, but with fixing of the temperature level on the surface at *T_w_* = 1850–1860 °C. It should be noted that after reaching the mode, the duration of the controlled isotherm for all samples was 210–220 s. Further, during the next 200–210 s, a spontaneous monotonic increase in the temperature of the samples was observed up to *T_w_* = 1970–1980 °C, after which the tests were stopped. Typical results of firing experiments are shown in Figure 5. From the presented photograph of the surface in Figure 5b, it can be seen that in the vicinity of the epicenter of the impact of the flow, there were traces of boiling and erosion of the coating. It is probable that the uncontrolled heating of the surface of the samples mainly occurred as a result of an increase in the catalyticity of the coating with respect to the reactions of heterogeneous recombination of atoms and ions of the air plasma (O, N, O^+^, N^+^, NO^+^) into molecules (O_2_, N_2_, NO) [13,23,29,37]. This, in turn, is associated with breaks in the continuity of the oxide film due to the formation and entrainment of volatile compounds (mainly SiO), the vapor pressure of which increases with increasing temperature [35]. Additionally, the possibility of influencing the temperature rise by a probable decrease in the emissivity of the surface as a result of an increase in the thickness of the oxide layer is also not excluded. In this case, the degree of blackness decreases from the values characteristic of a non-oxidized surface to the values inherent in oxide films—primarily, the glass phase composed of SiO_2_. The presence of areas with different colors (dark blue, light blue, gray) indicates a different degree of degradation of the coating with increasing distance from the place of localization of the plasma flow [23]. The average rate of mass loss in the samples for the entire time of the firing experiment in this mode was 13 ± 2 mg/(cm^2^·h).

The test results are summarized in Table 3 (positions No. 3, 4). When calculating the value of the specific weight loss, the value of the surface area of the windward side of the samples was used, and when evaluating the average rate of weight loss, only the time interval was taken into account, starting from the moment the temperature *T_w_* reached a given level and until the end of the test.

The base coating of the Si–TiSi_2_–MoSi_2_–TiB_2_ system, tested in similar firing experiments, completely loses its protective properties and operability when the surface temperature *T_w_* approaches ~ 1800 °C. Thus, under conditions of stepwise heating with an increase in the pressure *P*_0_ in the preheater chamber of the VAT-104 installation from 34 to 38 kPa, at 600 s from the start of the test, a rapid uncontrolled increase in temperature on the front surface of the sample was recorded from *T_w_* ~ 1750 °C to *T_w_* ~ 2130 °C (Figure 6a). Subsequent examination of the sample revealed on most of its front side (Figure 6b) the presence of traces of intense boiling of the coating with the formation of numerous bubbles and burnouts up to the carbon base (deep black cavities in the vicinity of the critical point). The rate of weight loss by the sample for the entire time of the firing experiment in this regime was 20.7 mg/(cm^2^·h).

To expand information on the behavior of the coating under study in the Si–TiSi_2_–MoSi_2_–TiB_2_–CaSi_2_ system at lower operating temperatures, a number of firing experiments were carried out with a stepwise increase in the pressure in the preheater *P*_0_ from 12 to 22 and from 22 to 34 kPa, providing heating of the samples in the interval *T_w_* = 1300–1500 and 1500–1750 °C, respectively. The exposure time of the samples at each temperature range was 1200 ± 5 s. The coating on all specimens remained intact. The results of the experiments are summarized in Table 3 (positions No. 1, 2). At *T_w_* = 1300–1500 °C, the weight gain of the samples was observed, which indicates the prevalence of the process of formation of the oxide film over the loss of mass as a result of erosion and sublimation.

The second series of tests was carried out with samples of a C_f_/C–SiC composite with a coating, located at a distance of 56 mm from the outlet section of the nozzle of the VAT-104 setup at a constant power of the anode power supply of the induction plasmatron *W_a_* = 211 ± 4 kW. A decrease in the distance to the nozzle exit (in contrast to the first series) led to an increase in the braking enthalpy. This was due to a reduction in the area of the disturbed air plasma flow and an increase in the steepness of the shock front, as well as an increase in the maximum temperature in the flow core (and, consequently, the gradient of the heat flux density along the jet diameter). Air plasma parameters were within the following limits: flow velocity 4.5–4.8 km/s; enthalpy of flow 45–50 MJ/kg; flow stagnation temperature ~ 10,000 K; the degree of air dissociation in the flow was 85–90%; the degree of ionization was about 1%.

Four samples were investigated under conditions of stepwise heating in the temperature range on the coating surface from *T_w_* = 1300 °C to *T_w_* = 1900–1920 °C with exposure at each stage for 120 s (exposure in the range *T_w_* = 1840–1870 °C was 180 s). The pressure *P*_0_ in the pre-heater chamber was changed from 10 to 25 kPa every 5 kPa, and then to 37.5 kPa every 2.5 kPa. The total time of each firing experiment was 1410 ± 10 s, the characteristic parameters of the regime are shown in Figure 7a. A typical appearance of the front surface of the samples after testing is shown in Figure 7b. The coating on all four samples retained its integrity and was characterized by a vitreous sheen. The color of the coating in the vicinity of the epicenter of the impact of the flow was bluish gray, whereas far from the epicenter it was light gray. Estimated average values of the specific weight loss of the samples and its rate at *T_w_* = 1900–1920 °C are presented in Table 3 (position No. 5). The average rate of mass loss by the samples for the entire time of the firing experiment under this regime was 31.4 ± 5.4 mg/(cm^2^·h).

It is known [9,13,36] that with an increase in the operating temperatures of boride-silicide coatings, including those containing Si and/or SiC, above *T_w_* = 1730–1750 °C, a sharp intensification of the processes of sublimation of the borosilicate glass phase occurs as a result of the formation of volatile oxides SiO and BO_x_, the vapor pressure of which increases with increasing temperature and decreasing external pressure. This, in turn, leads to disruption of the continuity of the oxide film and to additional heating as a result of an increase in catalyticity with respect to the reactions of heterogeneous recombination of atoms and ions of the air plasma at active centers of the surface. The passive mode of oxidation of coatings is replaced by an active one, which leads to a complete loss of their protective ability.

The nature of temperature curve 2, depending on the stagnation pressure 1 increasing stepwise in time (Figure 7a), indicated the prevalence of the rate of formation of a low-catalytic oxide film on the surface of the studied coating over its erosion and sublimation at *T_w_* > 1750 °C under conditions of significant external rarefaction. This was confirmed by the regular decrease in the steady-state radiation-equilibrium temperature of the surface of the samples 40–55 s after the transition to the next pressure step. Thus, the temperature value at the critical point decreased from *T_w_* = 1872 °C at 945 s of the experiment to *T_w_* = 1838 °C at 1067 s of testing at a constant value of *P*_0_ = 32.5 kPa; with *T_w_* = 1900 °C for 1150 s experiment to *T_w_* = 1877 °C for 1234 s testing at *P*_0_ = 35 kPa; and with *T_w_* = 1925 °C at 1318 from the test to *T_w_* = 1905 °C at 1412 s from the test at *P*_0_ = 37.5 kPa (see a fragment of the *T_w_*(τ) curve on an enlarged scale in the inset to Figure 7a). Regeneration and restoration of the continuity of the film composed of modified borosilicate glass for the specified period of time led to a decrease in the catalytic activity of the coating, and, consequently, to the “dropping” of the surface temperature.

It should also be noted that the possibility of influencing the temperature decrease by a probable increase in the emissivity of the oxide film, at the time of its discontinuity and an increase in porosity as a result of the release of gaseous oxidation products of the subsurface layers of the outside coating, is not excluded. This issue requires individual research and will be considered separately.

The other three samples were tested under conditions of stepwise heating in temperature ranges on the coating surface from *T_w_* = 1500 °C to *T_w_* = 1940–1960 °C, with holding at each stage for 120 s (holding in the interval *T_w_* = 1645–1665 °C was 300 s). The pressure *P*_0_ in the preheater chamber was set at 20 kPa, then increased to 25 kPa, and then to 32.5 kPa every 2.5 kPa. It should be noted that in the last seconds of the firing experiments in this mode, a spontaneous increase in the temperature of the surface of the samples in the vicinity of the edge up to *T_w_* = 2250–2270 °C was observed, after which the tests were stopped. The total time of each firing experiment was 920 ± 15 s; the characteristic parameters of the regime are shown in Figure 8a. A typical appearance of the front surface of the samples after testing is shown in Figure 8b. It can be seen that in the vicinity of the epicenter of the impact of the flow, the coating had an increased roughness with characteristic signs of boiling. In the area of the edge of the sample, where during the test there was a “burn-up”, the texture of the C_f_/C–SiC was observed, which indicated the complete absence of coating. The color of the coating in the vicinity of the epicenter of the impact of the flow was bluish gray, whereas far from the epicenter it waslight gray. The estimated average values of the specific weight loss of the samples and its rate at *T_w_* = 1940–1960 °C are presented in Table 3 (position No. 6). The average rate of mass loss by the samples for the entire time of the firing experiment in this regime was 81.1 ± 11.7 mg/(cm^2^·h).

Analysis of the thermal images of the surface of the samples and the temperature distribution over their diameter in the process of firing tests made it possible to conclude that in the experiments carried out according to the regime shown in Figure 8a, deviation of the core of the incident plasma flow from the axis of symmetry of the samples was observed. That led to a certain increase in heat density in the zone of the lateral edge. The misalignment of the aerogasdynamic flow and heating caused the samples to “burn out” from the edges.

In general, for both series of tests, good reproducibility of experimental data was established, indicating the identity of the physicochemical processes occurring in the studied coating during its interaction with air plasma, and small values of random errors. The temperatures of the surfaces of the samples measured by the pyrometer, taking into account corrections for the degree of emissivity of the coating ε = 0.7, are the lower estimates of their real values, since during operation the emissivity of the coating tends to decrease (the surface “turns gray”).

The processing of experimental data of gas-dynamic tests together with the results of parametric numerical modeling made it possible to obtain additional information on the catalytic activity of the coating in relation to the reactions of heterogeneous recombination of atoms and ions of air plasma. The obtained values of the rate constant of heterogeneous recombination *K_w_* are presented in Table 3. It should be noted that under similar test conditions for promising slip-firing coatings of the HfB_2_–SiC–HfO_2_–ZrO_2_–Y_2_O_3_ system [45], the *K_w_* values changed abruptly from 2.0 to 23.0 m/s when passing through the temperature *T_w_* ~ 1730 °C. This led to an increase in heat fluxes to the samples by a factor of 3–5 and, as a consequence, to an additional increase in temperatures *T_w_* by 600–1000 °C. Therefore, the coating presented in this work should be classified as low-quality, which gives it an additional advantage over coatings of other systems (especially over compositions based on super-refractory borides and zirconium and/or hafnium carbides).

### 3.5. Structural-Phase Studies of the Coating after Exposure to an Air Plasma Flow

Typical results of structural studies of cross-sectional sections of specimens coated with the Si–TiSi_2_–MoSi_2_–TiB_2_–CaSi_2_ system after gas-dynamic tests at *T_w_* = 1810–1820 °C are shown in Figure 9. The white and light gray patterns observed in the oxide zone should be attributed to artifacts of SEM studies of dielectrics with high surface resistivity, which include silicate glasses. It is seen that during the operation of the coating, a continuous oxide film of various thicknesses is formed on its surface. So, for the areas located in the vicinity of the epicenter of the impact of the flow, the thickness of the film is 40–50 microns (Figure 9a), while in the peripheral areas it is within 10–12 microns (Figure 9b). The thicknesses are indicated without taking into account the mechanical carryover of a part of the oxide film as a result of erosion upon interaction with the flow. The thickness of the main (unoxidized) coating layer, on the contrary, decreases as it approaches the epicenter of the flow due to the greater depletion of the structural components of the layer, as a result of the more intense formation and growth of the oxide film in comparison with the peripheral zones (10–15 μm in the central part of the samples in comparison to 30–35 microns when approaching the edges). The presence of an unoxidized layer indicates the preservation of the protective capabilities of the coating under conditions of fire tests, according to the regime shown in Figure 4a.

Separate X-ray maps of the distribution of elements in the cross-section of the coating (Figure 9c) and a multilayer combined image (Figure 9d), created on the basis of combining an electronic image and X-ray maps, clearly demonstrate the morphology of the structure of the marked layers.

A controlled increase in operating temperatures up to *T_w_* = 1850–1860 °C led to disruption of the continuity of the oxide film as a result of the formation and development of gas-filled cavities at the boundary “oxide film–main coating layer”. The coating went into boiling mode with further uncontrolled temperature rises and intensification of the structure degradation processes. Typical microstructures of the coating cross-sections for different areas are shown in Figure 10. In the vicinity of the epicenter of the effect of the flow, through oxidation of the main layer of the coating, an almost complete violation of the integrity of the oxide film was observed as a result of the breakthrough of gas-filled cavities (Figure 10a). The protective properties of the coating were exhausted. On the contrary, in the peripheral zones the presence of both an oxide film with a thickness of 25–30 µm with gas-filled bubbles in its lower part and an unoxidized coating layer with a thickness of 10–15 µm was noted (Figure 10b,c).

A detailed study of the cavities at the interface between the oxide film and the main coating layer made it possible to establish in them the in situ formation of fiber-like Si_2_ON_2_ aggregates (Figure 11) with a distinct core-shell structure containing nanocrystalline regions. The morphology of the Si_2_ON_2_ phase and its temperature resistance (*T_decomp_* > 1900 °C) ensured the realization of the effect of natural reinforcement of the oxide film, which increased its resistance to mechanical entrainment in flows. The formation of such fibers was established in [46,47].

Structural studies of the oxide film from the surface of the samples revealed the presence of the effect of high-temperature phase separation and confirmed the assumption of an increase in the degree of heterogeneity as a result of doping with calcium. The SEM and EDS results, partially shown in Figure 12, showed that the heterogeneous structure of the film was represented mainly by borosilicate glass, thin TiO_2_ needles oriented at different angles to the surface, as well as micro-inhomogeneous regions with a diameter of no more than 3–5 μm, containing O–Si–Ti–Ca elements in a ratio from 5:2:0:1 to 5:1:1:1. Based on the results of the oxidation of compacts under static conditions (Section 3.3) and the phase diagram in the CaO–TiO_2_–SiO_2_ system [39], it can be assumed that these regions are represented by solutions in the CaSi_2_O_5_–CaTiSiO_5_ system immiscible with the glass phase based on SiO_2_.

The results of the XRD analysis performed from the surface of the samples after the firing experiments were in good qualitative agreement with the SEM and EDS data. In addition to the set of reflections characteristic of the coating in the initial state (Table 1), a wide halo at 2θ = 20–24° corresponding to amorphous SiO_2_ (JCPDS card No. 29-0085) and peaks related to the crystalline phases of SiO_2_ (JCPDS card No. 39-1425) and TiO_2_ (JCPDS card No. 21-1276) in the form of cristobalite and rutile, respectively, were observed. A feature of the obtained X-ray diffraction patterns was the revealed crystallographic texture of the TiO_2_ phase, mainly in the [1 1 0] direction. Traces of titanite CaTiSiO_5_ (JCPDS card No. 25-0177) were also recorded in the oxide layer. Quantitative XRD analysis was impossible due to the different thickness of the formed oxide film and the change in the proportion of the amorphous and crystalline components of the SiO_2_ phase with increased distance from the epicenter of the flow, as well as due to differences in the roughness of the surface areas and revealed texture phenomena.

### 3.6. Serviceability and Degradation of the Coating Structure in High-Speed Air Plasma Flows

An analysis of the results obtained in this work, together with the results of earlier studies [11,23,24,40], made it possible to establish the following. Under the conditions of interaction of the coating of the Si–TiSi_2_–MoSi_2_–TiB_2_–CaSi_2_ system with a high-speed air plasma flow, at temperatures up to *T_w_* = 1810–1820 °C, the surface is passivated, which prevents the development of active oxidation further into the coating. The oxidation processes of the coating components are described by the reactions:Si + O_2_↑ → SiO_2_;(3)
5MeSi_2_ + 7O_2_↑ → 7SiO_2_ + Me_5_Si_3_ (Me–Mo, Ti*_x_*Mo_1−*x*_);(4)
TiSi_2_ + 3O_2_↑ → 2SiO_2_ + TiO_2_;(5)
2CaSi_2_ + 5O_2_↑ → 4SiO_2_ + 2CaO;(6)
2TiB_2_ + 5O_2_↑ → 2TiO_2_ + 2B_2_O_3_;(7)
SiC + 2O_2_↑ → SiO_2_ + CO_2_↑;(8)
2SiC + 3O_2_↑ → 2SiO_2_ + 2CO↑;(9)
and are accompanied by the interaction of oxidation products:*x*SiO_2_ + *y*B_2_O_3_ → *x*SiO_2_·*y*B_2_O_3_;(10)
2SiO_2_ + CaO → CaSi_2_O_5_;(11)
SiO_2_ + CaO + TiO_2_ → CaTiSiO_5_.(12)

The formation of a protective film based on SiO_2_ occurs during the oxidation of both free silicon, which is part of the eutectic of the main coating layer, and higher silicides MeSi_2_ (Ti_x_Mo_1−x_Si_2_, TiSi_2_, MoSi_2_, CaSi_2_) and SiC carbide, forming a branched framework (reactions (3)–(6) and (8), (9), respectively). Oxidation of titanium-containing phases TiSi_2_ and TiB_2_ leads to the formation of needle-shaped crystals of TiO_2_ in the form of rutile (reactions (5), (7)), which are immiscible with silica. The oriented arrangement of TiO_2_ crystals leads to the appearance of the effect of “reinforcement” of the oxide film, and, consequently, to an additional increase in its resistance to mechanical entrainment in flows.

Simultaneously, with oxidation and the formation of a continuous outer oxide layer, segregation of calcium in its outer part occurs—initially uniformly distributed in the coating volume, apparently due to a high diffusion rate in viscous-plastic components (in eutectic and glass phase). Segregation of calcium is associated, on the one hand, with its surface activity in relation to silicon and titanium—characterized by a lower value of surface tension. On the other hand, calcium has a higher reactivity when interacting with oxygen than titanium, silicon and boron, as evidenced by comparative data on the differences in the values of relative electronegativities according to L.K. Pauling between oxygen and these elements.

The interaction of primary oxidation products with each other, limited by the rate of diffusion mass transfer of reagents in a complex heterogeneous system, leads to the formation of: (i) borosilicate glass phase with a single anionic matrix according to reaction (10); and (ii) new crystalline phases CaSi_2_O_5_ and CaTiSiO_5_ according to reactions (11) and (12), respectively. In this work, the distribution of boron in glass was not quantitatively estimated due to the low sensitivity of the EDS to light elements, especially with a low content (in reaction (10) y << x). The formation of solutions in the CaSi_2_O_5_–CaTiSiO_5_ system that are immiscible with the borosilicate glass phase confirmed the effect of liquid-phase separation in the SiO_2_–TiO_2_–CaO system, which was reported in [34,39,40].

An increase in the operating temperatures on the surface of up to *T_w_* ~ 1850–1860 °C and a slowdown in the diffusion of oxygen into the coating volume through the growing heterogeneous oxide layer led to the interaction of free silicon and SiC with the SiO_2_ film at its inner boundary, and to the formation of gaseous products underneath the oxide film (at the boundaries of Si–SiO_2_ and SiC–SiO_2_) according to the reactions:Si + SiO_2_ → 2SiO↑ (ΔG < 0 kJ/mol at *T* > 1872 °C);(13)
SiC + 2SiO_2_ → 3SiO↑ + CO↑ (ΔG < 0 kJ/mol at *T* > 1883 °C).(14)

As a result, cavities filled with gases, mainly silicon monoxide SiO, were formed under the oxide film (Figure 10b,c). Moreover, the process of formation of SiO will be more intense, the lower the partial pressure of oxygen in the external gas environment [23,35]. The interaction of volatile silicon monoxide SiO with O_2_ and N_2_ diffusing through a heterogeneous oxide film leads to an increase in the thickness of the SiO_2_ film from its lower boundary according to reaction (15) and in situ synthesis of silicon oxynitride Si_2_ON_2_ fibers according to reaction (16):2SiO↑ + O_2_↑ → 2SiO_2_;(15)
4SiO↑ + 2N_2_↑ → 2Si_2_ON_2_ + O_2_↑.(16)

Reaction (15) makes a positive contribution to the compensation of SiO_2_ losses due to erosive entrainment from a surface streamlined by high-speed flows. The formation of Si_2_ON_2_ fibers in the oxide film contributes to its additional reinforcement, which increases the resistance to mechanical entrainment (erosion).

As the test duration increases at the indicated temperatures, gases accumulate under the oxide film, gas-filled cavities grow, and gaseous products penetrate the viscous borosilicate glass. The continuity of the oxide film is violated and intensive oxidation of free silicon and SiC begins according to the reactions:2Si + O_2_↑ → 2SiO↑;(17)
SiC + O_2_↑ → SiO↑ + CO↑.(18)

That is, the oxidation of the coating goes into an active stage. Abundant blistering of the coating is observed (the “boiling” effect), leading to a breakdown degradation of the oxide film (the effect of “cracking” from the inside; Figure 10a). The absence of a film based on amorphous silica or its discontinuity, in turn, leads to an increase in the catalyticity of the coating surface with respect to reactions of heterogeneous recombination of atoms and ions of air plasma, and, consequently, to additional heating of the surface [37]. This, in turn, contributes to the intensification of the above-described processes of degradation of the coating structure and to the complete loss of its performance.

It should be noted that chemical reactions (3)–(9) and (15), (17) and (18) were recorded with the participation of molecular oxygen. The latter has a significantly lower reactivity than atomic oxygen. The presence of a predominant amount of atomic oxygen in the air plasma flow (air undergoes dissociation in a shock wave in front of the samples) leads to an acceleration of oxidative processes and an increase in the diffusion permeability of the oxidizer through the growing oxide layer.

Thus, it was experimentally established that a simultaneous increase in the fraction of the refractory TiB_2_ phase in the coating of the Si–TiSi_2_–MoSi_2_–TiB_2_–CaSi_2_ system and the degree of heterogeneity of oxide films formed during its oxidation led to an expansion of the temperature-time ranges of the protective action of the coating, in comparison with the coatings of the basic Si–TiSi_2_–MoSi_2_–TiB_2_–CaSi_2_ system. Thus, the performance of the coating of the Si–TiSi_2_–MoSi_2_–TiB_2_–CaSi_2_ system under the studied conditions of interaction with plasma flows was at least 920–930 s at *T_w_* = 1810–1820 °C, at least 510–520 s at *T_w_* = 1850–1860 °C, not less than 280–290 s at T_w_ = 1900–1920 °C, and not less than 100–110 s at *T_w_* = 1940–1960 °C. Under similar conditions, the coating of the Si–TiSi_2_–MoSi_2_–TiB_2_–CaSi_2_ system is efficient for no more than 50 s at *T_w_* ≥ 1800 °C. The results obtained in combination with the data of [11,23,24,40] confirm the correctness of the chosen direction of improving the coatings of the Si–TiSi_2_–MoSi_2_–TiB_2_–CaSi_2_ system and determine the need for parametric studies in the development of optimal concentration boundaries of compositions in the Si–TiSi_2_–MoSi_2_–TiB_2_–CaSi_2_ system.

## 4. Conclusions


Systematic research aimed at the development and improvement of heat-resistant self-healing coatings based on silicide–boride systems has been continued. The coatings are designed to protect against high-temperature oxidation of carbon–carbon and carbon–ceramic composite materials, which are promising for use in thermal protection systems for airframes and flow paths of propulsion systems of atmospheric high-speed aircraft and reusable aerospace vehicles.A heterophase powder material of experimental composition in the Si–TiSi_2_–MoSi_2_–TiB_2_–CaSi_2_ system was obtained by induction melting in a suspended state, followed by dispersion in a rotating ball mill. The phase composition of the powder included, wt%: 39.6 Ti_0_._8_Mo_0_._2_Si_2_, 33.6 TiB_2_, 17.8 CaSi_2_, 4.5 TiSi_2_, and 4.5 MoSi_2_. The powder was consolidated by hot pressing. The kinetics of static oxidation of compacts in air under thermal cycling conditions at 20 °C ↔ 1650 °C for 60 min was investigated. The total number of thermal cycles was 5. The oxidation process of compacts obeyed the power law with an exponent *n* = 2.695, which indicates a significant effect on the course of the process of evolutionary changes in the structure of the formed oxide film.Using the technology of firing deposition of layers, a thin-layer coating was formed on samples from a C_f_/C–SiC composite. The resulting powder, dosed with silicon in a mass ratio of 4: 1, was used as a starting material. Firing was carried out to a temperature of 1480 ± 5 °C with a residual pressure in the chamber of 8–9 MPa.Fire gas-dynamic tests of samples from a C_f_/C–SiC composite with a coating of the Si–TiSi_2_–MoSi_2_–TiB_2_–CaSi_2_ system was carried out under conditions of aerogasdynamic flow and non-equilibrium heating by air plasma flows at Mach numbers M = 5.5–6.0 and enthalpy 40–50 MJ/kg. The coating remained operational at *T_w_* = 1810–1820 °C for at least 920–930 s, at *T_w_* = 1850–1860 °C for at least 510–520 s, at *T_w_* = 1900–1920 °C for at least 280–290 s, and at *T_w_* = 1940–1960 °C for not less than 100–110 s. The average values of the rate of loss of the coating mass in the indicated temperature ranges *T_w_* were, mg/(cm^2^·h): 30.1 ± 3.3, 83.5 ± 12.8, 145.3 ± 24.7, and 494.9 ± 71.8, respectively.The rate constant of heterogeneous recombination *K_w_* of atoms and ions of the air plasma at the active centers of the coating surface was estimated. The average *K_w_* values in the above temperature ranges *T_w_* were m/s: 4.7 ± 1.3, 6.5 ± 1.5, 8.0 ± 2.0, and 9.0 ± 3.0, respectively.It has been established that the performance of the coating was provided by the structural-phase state of its main layer, the formation and evolution on its surface during operation of a passivating heterogeneous oxide film composed of borosilicate glass with liquation inhomogeneities in titanium and calcium and reinforcing microneedles of titanium oxide in the form of rutile, and in situ by fibers of silicon oxynitride Si_2_ON_2_. A decrease in the vapor pressure in the “oxide film–coating” system with an increase in the degree of heterogeneity of the outer layer was experimentally confirmed. At *T_w_* ≥ 1850–1860 °C, intense gas formation was observed at the “oxide layer–coating” interface as a result of the generation of volatile silicon monoxide with the subsequent realization of the effects of boiling and breakdown degradation of the oxide film.


## Figures and Tables

**Figure 1 nanomaterials-11-02637-f001:**
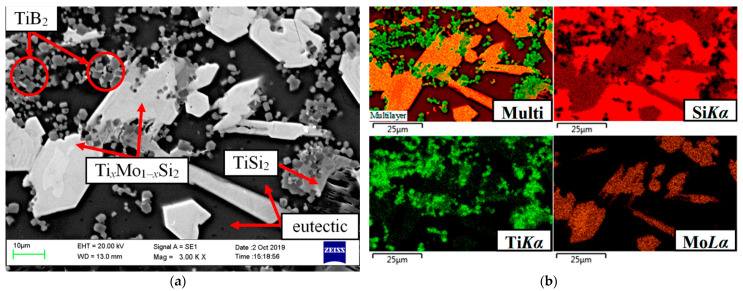
Microstructure of the coating in the initial state (**a**) and the distribution of elements in characteristic X-ray radiation (**b**).

**Figure 2 nanomaterials-11-02637-f002:**
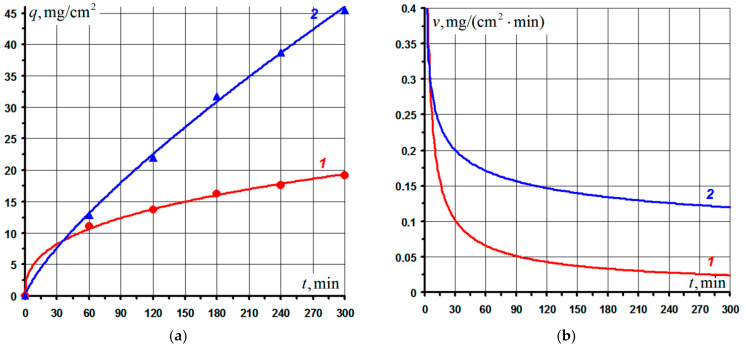
Kinetic curves of the oxidation of compacts from powders of experimental composition in the Si–TiSi2–MoSi2–TiB2–CaSi2 system (1) and basic composition in the Si–TiSi2–MoSi2–TiB2 system (2) at 1650 °C in air (**a**) and their corresponding oxidation rates (**b**).

**Figure 3 nanomaterials-11-02637-f003:**
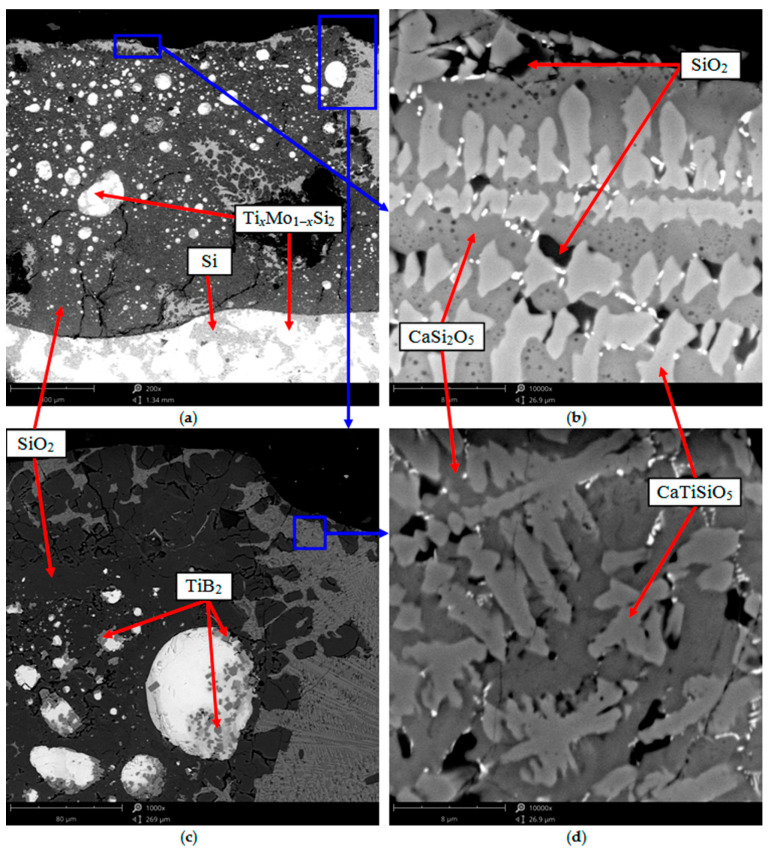
Microstructure of a thin section of a compact of composition 1 in the area of the surface layer after 5 h isothermal oxidation at 1650 °C in air: (**a**) ×200; (**b**) ×10,000; (**c**) ×1000; (**d**) ×10,000.

**Figure 4 nanomaterials-11-02637-f004:**
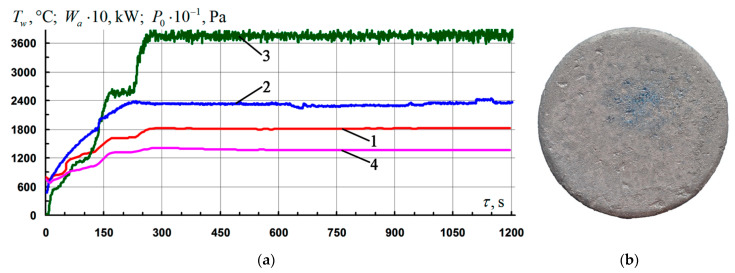
Results of gas-dynamic tests of C_f_/C–SiC samples coated with the Si–TiSi_2_–MoSi_2_–TiB_2_–CaSi_2_ system at *T_w_* = 1810–1820 °C: (**a**) change of mode parameters in time: 1, 4—temperature of the front/rear surface of the sample at the critical point, *T_w_*; 2—generator power, *W_a_*; 3—braking pressure in the pre-heater chamber, *P*_0_; (**b**) a typical view of the front side of the samples after testing.

**Figure 5 nanomaterials-11-02637-f005:**
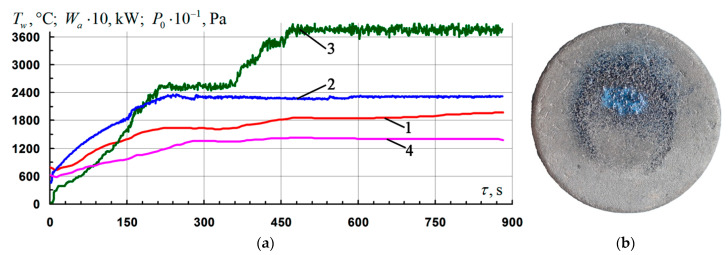
Results of gas-dynamic tests of C_f_/C–SiC samples coated with the Si–TiSi_2_–MoSi_2_–TiB_2_–CaSi_2_ system at *T_w_* = 1850–1860 °C: (**a**) change of mode parameters in time: 1, 4—temperature of the front/back surface of the sample at the critical point, *T_w_*; 2—generator power, *W_a_*; 3—braking pressure in the pre-chamber of the heater, *P*_0_; (**b**) typical view of the front side of the samples after testing.

**Figure 6 nanomaterials-11-02637-f006:**
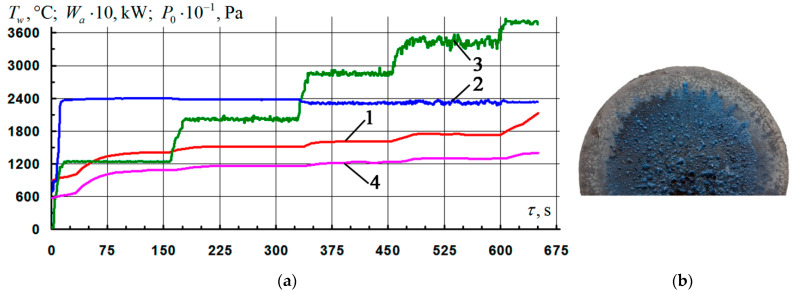
Results of gas-dynamic tests of C_f_/C–SiC samples with a base coating of the Si–TiSi_2_–MoSi_2_–TiB_2_ system at *T_w_* ~ 1800 °C: (**a**) change of mode parameters in time: 1, 4—temperature of the front/back surface of the sample at the critical point, *T_w_*; 2—generator power, *W_a_*; 3—braking pressure in the pre-chamber of the heater, *P*_0_; (**b**) typical view of the front side of the samples after testing.

**Figure 7 nanomaterials-11-02637-f007:**
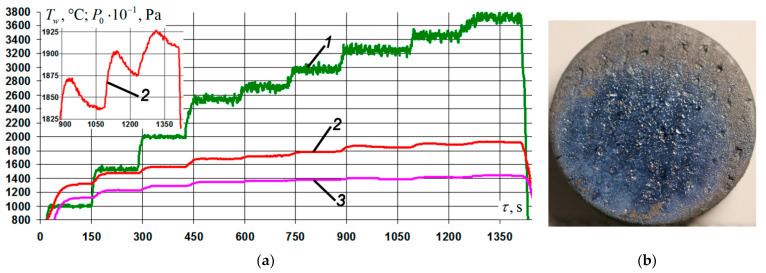
Results of gas-dynamic tests of C_f_/C–SiC samples coated with the Si–TiSi_2_–MoSi_2_–TiB_2_–CaSi_2_ system at *P*_0_ = 10–37.5 kPa: (**a**) change of mode parameters in time: 1—braking pressure in the pre-chamber of the heater, *P*_0_; 2, 3—temperature of the front/back surface of the sample at the critical point, *T_w_*; (**b**) typical view of the front side of the samples after testing.

**Figure 8 nanomaterials-11-02637-f008:**
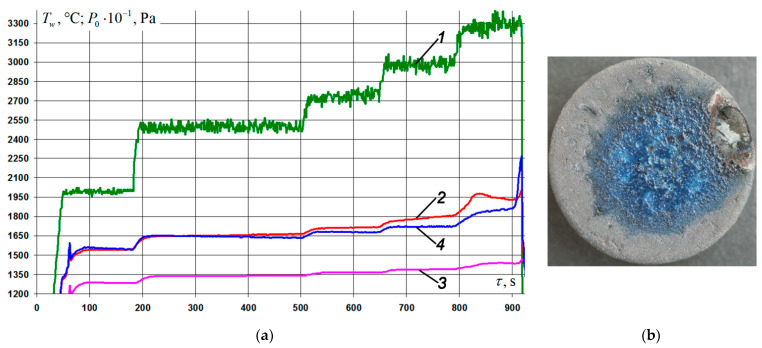
Results of gas-dynamic tests of C_f_/C–SiC samples coated with the Si–TiSi_2_–MoSi_2_–TiB_2_–CaSi_2_ system at *P*_0_ = 20–32.5 kPa: (**a**) change of mode parameters in time: 1—braking pressure in the pre-chamber of the heater, *P*_0_; 2, 3—temperature of the front/back surface of the sample at the critical point, *T_w_*; 4—temperature of the front surface of the sample in the “heat” zone, *T_w_*; (**b**) typical view of the front side of the samples after testing.

**Figure 9 nanomaterials-11-02637-f009:**
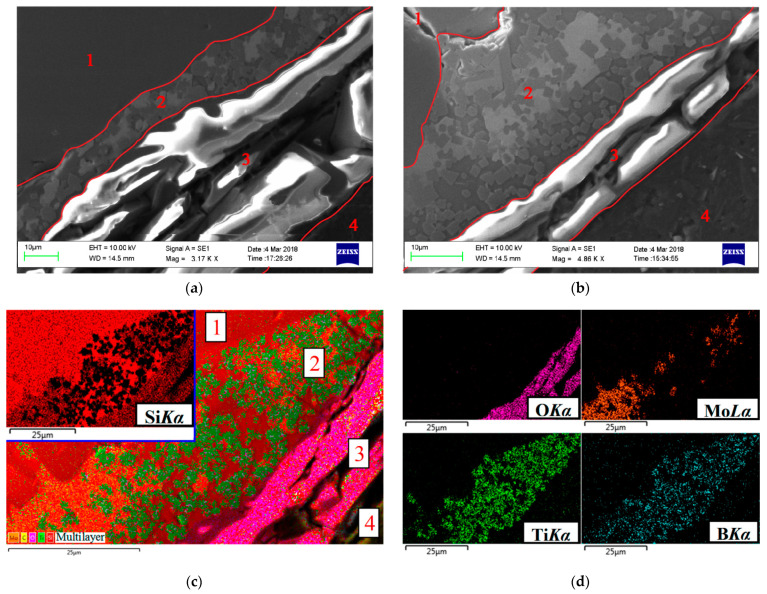
Microstructure (**a**,**b**) and distribution of elements in characteristic X-ray radiation (**c**,**d**) in cross-sections of Si–TiSi_2_–MoSi_2_–TiB_2_–CaSi_2_ coating areas after firing tests according to the regime shown in Figure 4a: (**a**) in the vicinity of the epicenter of the flow; (**b**) at a distance of 10 mm from the critical point; (**c**,**d**) at a distance of 7.5 mm from the critical point: 1—SiC-backing layer, 2—unoxidized coating, 3—oxide film, 4—resin.

**Figure 10 nanomaterials-11-02637-f010:**
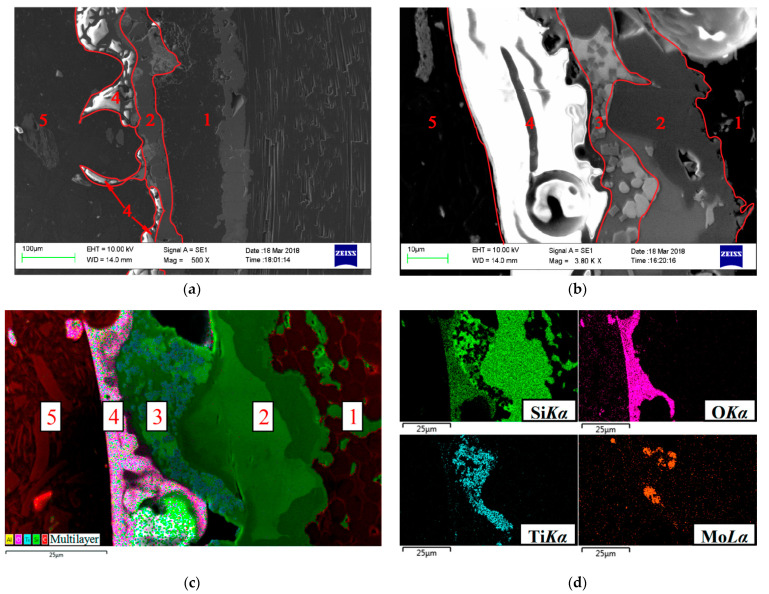
Microstructure (**a**,**b**) and distribution of elements in characteristic X-ray radiation (**c**,**d**) in cross-sections of Si–TiSi_2_–MoSi_2_–TiB_2_–CaSi_2_ coating areas after firing tests according to the regime shown in Figure 5, a: (**a**) in the vicinity –of the epicenter of the flow; (**b**) at a distance of 10 mm from the critical point; (**c**,**d**) at a distance of 7.5 mm from the critical point: 1—substrate, 2—SiC-backing layer, 3—unoxidized coating, 4—oxide film, 5—resin.

**Figure 11 nanomaterials-11-02637-f011:**
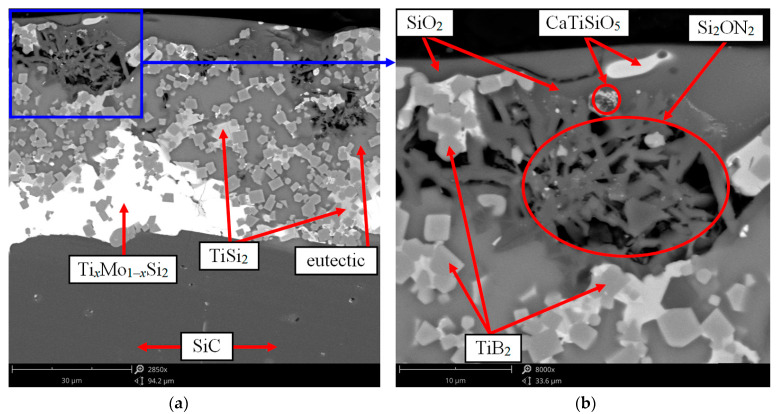
The microstructure of the cross section of the coating containing in situ-formed Si_2_ON_2_ fibers after gas-dynamic tests according to the regime shown in Figure 7a: (**a**) ×2850; (**b**) ×8000.

**Figure 12 nanomaterials-11-02637-f012:**
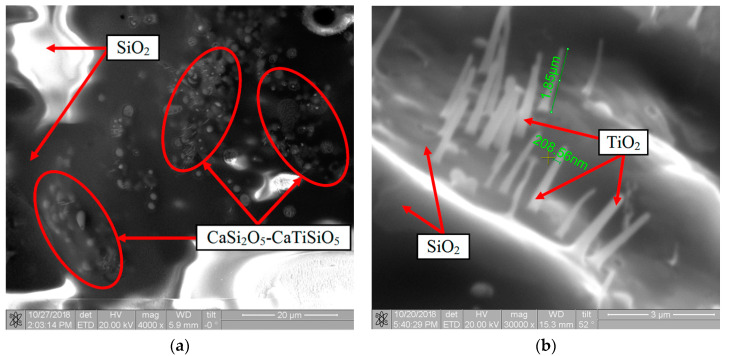
The microstructure of the surface of the oxide film on the Si–TiSi_2_–MoSi_2_–TiB_2_–CaSi_2_ coating after gas-dynamic tests according to the regime shown in Figure 4a: (**a**) ×4000; (**b**) ×30,000.

**Table 1 nanomaterials-11-02637-t001:** Phase composition of the obtained powder and coating.

Share	TiB_2_ (*P*6/*mmm*)	Ti_0.8_Mo_0.2_Si_2_ (*P*6_2_22)	TiSi_2_ (*Fddd*)	MoSi_2_ (*I*4/*mmm*)	CaSi_2_ (41/*amd*)	CaSi_2_ (*R*3*m*)	Si (*Fd*-3*m*)	SiC (*P*63*mc*)
JCPDS card No.	010-85-2083	07-0331	33-1384	030-65-2645	010-75-2193	010-75-2192	26-1481	010-72-0018
Powder XRD								
wt%	33.6	39.6	4.5	4.5	6.5	11.3	-	-
Cell parameters, Å	a = 3.031 c = 3.229	a = 4.688 c = 6.508	a = 8.262 b = 4.795 c = 8.546	a = 3.204 c = 7.845	a = 4.283 c = 13.52	a = 3.855 c = 30.6	-	-
Coating XRD								
wt%	26.8	31.9	3.6	3.2	5.2	8.9	19.1	1.3
Cell parameters, Å	a = 3.029 c = 3.226	a = 4.698 c = 6.523	a = 8.267 b = 4.797 c = 8.548	a = 3.202 c = 7.841	a = 4.282 c = 13.518	a = 3.857 c = 30.602	a = 5.428	a = 3.081 c = 15.123

**Table 2 nanomaterials-11-02637-t002:** Characteristics of the kinetics of compact oxidation at 1650 °C after total isothermal holding for 5 h.

Composition	System	*q*_Σ_, mg/cm^2^	*k*	*n*	*r*	*t*_1/2_, min	*v_av_*, mg/(cm^2^∙min)	
							Stage I	Stage II
*1*	Si–TiSi_2_–MoSi_2_–TiB_2_–CaSi_2_	19.2	9.782	2.695	0.998	46.3	0.209	0.038
*2*	Si–TiSi_2_–MoSi_2_–TiB_2_	45.5	0.457	1.286	0.999	123.1	0.187	0.130

**Table 3 nanomaterials-11-02637-t003:** Results of gas-dynamic tests of C_f_/C–SiC samples with a coating of the Si–TiSi_2_–MoSi_2_–TiB_2_–CaSi_2_ system at different modes.

№	Number of Samples	Test Mode	Flow Speed, km/s	Working Temperature *T_w_*, °C	Average Specific Weight Loss, mg/cm^2^	Average Rate of Weight Loss, mg/(cm^2^·h)	Recombination Rate *K_w_*, m/s
1	3	*P*_0_ = 12–22 kPa	4.3–4.5	1300–1500	+(1.8 ± 0.3)	+(5.4 ± 0.9)	1.0 ± 0.5
2	4	*P*_0_ = 22–34 kPa	4.3–4.5	1500–1750	0.7 ± 0.2	2.1 ± 0.6	2.0 ± 1.0
3	3	Figure 4a	4.3–4.5	1810–1820	7.8 ± 0.8	30.1 ± 3.3	4.7 ± 1.3
4	3	Figure 5a	4.3–4.5	1850–1860	10.0 ± 1.5	83.5 ± 12.8	6.5 ± 1.5
5	4	Figure 7a	4.5–4.8	1900–1920	12.3 ± 2.2	145.3 ± 24.7	8.0 ± 2.0
6	3	Figure 8a	4.5–4.8	1940–1960	20.7 ± 3.0	494.9 ± 71.8	9.0 ± 3.0

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
