# Peer review of "Oxidation Resistance of a Si–TiSi2–MoSi2–TiB2–CaSi2 Coating on a Cf/C–SiC Substrate in High-Speed High-Enthalpy Air Plasma Flows"

_nanomaterials, 2021, doi:10.3390/nano11102637_

Round 1

Reviewer 1 Report

The manuscript presents an investigation on a new protecting layer for the CfC-SiC substrate in high speed high-enthalpy air plasma flow.

Even thought the layer is titanium based, the work appears as a sequel of other previous work using layer with same structure.

The work has moderate interest, however can be considered for publication upon major revision.

Please se my attached comments

Author Response

Dear reviewer!

An article «Oxidation resistance of the Si-TiSi2-MoSi2-TiB2-CaSi2 coating on a Cf/C-SiC substrate in high-speed high-enthalpy air plasma flows» was refined according to your comments. The changes are highlighting.

Reviewer 2 Report

The present authors have designed a complex coating, i.e Si-TiSi2-MoSi2-TiB2-CaSi2 on C/C-SiC composites, and measured the oxidation resistance in a high-temperature and high-speed flows. The results were analyzed systematically. The manuscript was logically good. I suggest the manuscript should be accepted after minor revision.

(1) Several protective coatings have been reported before, including MoSi2,ZrB2-SiC. The present manuscript discussed a coating containing five components. Have the author considered the reliability for the future application?

(2) The peeling property of the coating should be clarify in the manuscript.

(3) How the author design the coating system? please clarify in the introduction.

Author Response

(The authors gave the same response as above.)

Round 2

Reviewer 1 Report

Thanks  to have considered my comments The manuscript can be published now

Author Response

Dear reviewer!

An article «Oxidation resistance of the Si-TiSi2-MoSi2-TiB2-CaSi2 coating on a Cf/C-SiC substrate in high-speed high-enthalpy air plasma flows» was refined according to your comments. Minor spelling and punctuation changes have been made in the text of the article: lines 96, 128, 293, 314, 403, 768, 771.
